# Identification, Microhabitat, and Ecological Niche Prediction of Two Promising Native Parasitoids of *Tuta absoluta* in Kenya

**DOI:** 10.3390/insects13060496

**Published:** 2022-05-25

**Authors:** Sahadatou Mama Sambo, Shepard Ndlela, Hannalene du Plessis, Francis Obala, Samira Abuelgasim Mohamed

**Affiliations:** 1International Centre of Insect Physiology and Ecology (icipe), Nairobi P.O. Box 30772-00100, Kenya; sndlela@icipe.org (S.N.); fobala@icipe.org (F.O.); 2Unit for Environmental Sciences and Management, North-West University, Potchefstroom 2520, South Africa; hannalene.duplessis@nwu.ac.za

**Keywords:** indigenous parasitoids, molecular identification, morphological identification, South American tomato pinworm, parasitism rate, agroecology, habitat suitability

## Abstract

**Simple Summary:**

Since the arrival of *Tuta absoluta*, a multivoltine insect species whose larvae develop in leaves, fruits, flowers, buds, and stems of tomatoes, producers are facing one of its biggest production challenges. The pest continues to invade new areas, causing heavy losses in the tomato value chain. Sprays of synthetic insecticides have shown very low efficacy on this pest because of its inclination to develop resistance to various insecticide-active ingredients. Biological control is one of the most promising solutions for the management of this pest. In this work, we investigated the most efficient indigenous parasitoids associated with *T. absoluta* in Kenya and their preferable habitat and ecological niche suitability. We identified two species, *Stenomesius* sp. near *japonicus* and *Bracon nigricans*, with up to 17% and 21% parasitism respectively. *Stenomesius* sp. near *japonicus* was more abundant in greenhouses and non-insecticide-treated tomatoes while *B. nigricans* was more abundant in the field tomatoes with a low abundance of *Nesidiocoris tenuis.* The ecological niche of these two species showed that *B. nigricans* was suitable for establishment in sub-Saharan Africa, a big part of South America, and Australia in both current and future scenarios.

**Abstract:**

Associations between the South American tomato pinworm, *Tuta absoluta* (Meyrick) (Lepidoptera: Gelechiidae), and its native parasitoids need to be updated to increase the implementation of pest control strategies. In this study, *T. absoluta*-infested tomato plants were collected from three regions in Kenya. The emerged parasitoids were identified, and their abundance was correlated with agroecological parameters, *viz*. cropping systems, and the abundance of the predator *Nesidiocoris tenuis* Reuter (Hemiptera: Miridae). The study further conducted a habitat suitability prediction for the identified parasitoids. Two parasitoid species, *Bracon nigricans* (Szépligeti) (Hymenoptera: Braconidae) and *Stenomesius* sp. near *japonicus* (Ashmead) (Hymenoptera: Eulophidae) emerged from *T. absoluta* immature stages, with parasitism rates ranging from 0 to 21% and 0 to 17% respectively. Insecticide application and open field cropping negatively influenced the parasitism by *S.* sp. nr *japonicus*. Low occurrence of *N. tenuis* positively affected *B. nigricans* parasitism. The predicted occurrence of parasitoid species indicated vast suitable areas for *B. nigricans* in sub-Saharan Africa, Australia, and South America. Low suitability was observed for *S.* sp. nr *japonicus* in Africa. Therefore, native parasitoids, especially *B. nigricans* could be considered for implementation as a biocontrol agent in the Integrated Pest Management program of *T. absoluta*.

## 1. Introduction

Invasive species are known to exert undesirable effects on biodiversity and human health [1,2,3]. They also adversely impact economic activities and food security [4,5]. The spread, establishment, and devastating effects of *Tuta absoluta* (Meyrick) (Lepidoptera: Gelechiidae) have contributed to the over-reliance on synthetic chemical insecticides for the management of this pest in Africa [6,7]. The preference by farmers for chemical control can be attributed to the observable quick knockdown of pests after application. As a result, the adverse effects associated with the use of synthetic chemicals are therefore ignored [8,9]. Mahugija et al. [10] reported that 50% of tomatoes grown and sold in Tanzania, had pesticide residues exceeding the official maximum residue limits, indicating a higher risk for public health. At the forefront of health risks are tomato farmers who due to the *T. absoluta* menace, apply insecticides regularly, often without any personal protective equipment. For example, up to 60% of Kenyan farmers do not use the necessary protective clothing when applying pesticides and as a result, 26% of them are reported to be suffering from pesticide-related health problems [11]. Due to the high pesticide cost in Kenya, farmers often use pesticides bought from informal markets in neighboring countries (Uganda and Tanzania), which are not registered for use on any specific crop [12]. The use of synthetic insecticides also influences the natural assemblage of parasitoids and predators of the pest [8,13].

Biological control is recommended for the control of *T. absoluta* albeit integrated with other control methods [7,14]. In most cases, invasive pests often arrive in new regions with no co-evolved natural enemies. However, if effective native parasitoids are present in the agroecological system, new associations with the pest may be formed based on multi-level interactions [15]. When the prey/host population levels are low, parasitoids and predators can switch their host and prey preferences depending on the availability of alternatives [16,17,18,19]. The use of natural enemies has several advantages for environmental conservation, especially due to their non-effect on non-target organisms [20,21].

Native parasitoids and predators have been used for *T. absoluta* control in several regions around the world. For instance, an estimated 65% predation and 20% parasitism by native species were reported from Israel in open tomato fields where synthetic chemicals were not applied [22]. *Trichogramma* spp. are lucratively used in many countries [23]. For instance, in Spain, the release of 30 *Trichogramma achaeae* Nagaraja and Nagarkatti (Hymenoptera: Trichogrammatidae) adults per plant reduced *T. absoluta* damage by 91.74% under greenhouse conditions [24]. Thus, the identification of effective indigenous parasitoids associated with *T. absoluta* is useful in the implementation of successful biocontrol programs. 

Numerous parasitoids attacking *T. absoluta* in Africa have been described. These include the braconids *Cotesia vestalis* (Haliday), *Apanteles litae* Nixon, *Meteorus laphygmarum* Brues, *Chelonus sp.,* and *Diadegma insulare* (Cresson), the Ichneumonidae *Pristomerus pallidus* (Kriechbaumer) and the Eulophidae *Elasmus* sp. in Senegal [25]. In North Africa, the Eulophid, *Necrenus artynes* (Walker) and *Neochrysocharis formosa* (Westwood)*,* the Trichrommatid, *Trichrogramma bourarachae* Pintureau & Babault, the Patygastrid, *Telenomus* sp., and the Torymid, *Hemiptarsenus zilahisebessi* Erdös were discovered parasitizing *T. absoluta* under natural conditions [26,27,28]. Other native parasitoids of *T. absoluta* such as *Bracon nigricans* Szépligeti, *Bracon hebetor* (Say), *Ecdamua cadenat* (Risbec) (Hymenoptera: Torymidae), *Dolichogenidea appellator* (Telenga) (Hymenoptera: Braconidae), and *Neochrysocharis formosa* (Westwood) (Hymenoptera: Eulophidae) have been reported in Sudan [29,30]. In the Middle East and North Africa (MENA) region, there have been several efforts of parasitoid releases against *T. absoluta* [31], and this showed a significant reduction in the pest damage [32,33,34,35,36].

The assemblage of native parasitoids in various tomato-producing areas needs to be documented if they are to be incorporated into augmentative and conservative biological control approaches for the management of *T. absoluta.* Furthermore, habitat and climate are significant factors limiting the distribution and abundance of insect pests, as well as their physiology and reproduction [37,38,39,40,41,42]. Parasitoids and predator distribution on a local scale, are also affected by agronomic practices such as pesticide application [43,44]. It is against this background that the current study sought to assess the presence and distribution of parasitoid species in open-field and greenhouse tomatoes in Kenya and their effectiveness in controlling *T. absoluta* as well as to determine their suitable habitat for perseverance in biological control programs.

## 2. Materials and Methods

### 2.1. Parasitoid Collection in the Study Area

The investigation was conducted in Kirinyaga, Nakuru, and Nairobi Counties, Kenya with samples collected 12 times between March 2020 and October 2021. Collections were done six times in Kirinyaga and on three occasions in Nakuru and Nairobi. Sampling sites within the Counties were randomly selected based on the presence of tomato production farms with *T. absoluta* infestations in open fields and greenhouses. For each collection in a site, infested leaves were randomly picked and placed in transparent 4-liter plastic containers and labeled according to GPS coordinates and date of collection. Depending on the availability of infested plants, three to twelve samples were collected. The containers were closed with a mesh-infused lid for aeration and transported to the laboratory at *icipe.* The samples were weighed and incubated separately under ambient laboratory conditions (25 ± 1 °C; 65 ± 5% r.h; 12HL:12HD photoperiod). Un-infested tomato plants grown under standard agronomic practices in a greenhouse at *icipe* were added to the containers regularly as food for the larvae. The plastic containers were monitored daily, and pupae were transferred into Perspex cages (30 cm × 30 cm × 30 cm). The number of moths, parasitoids, and *Nesidiocoris tenuis* Reuter (Hemiptera: Miridae) that emerged from each sample were recorded. The parasitoids were grouped based on morphological similarities, counted, and transferred into Perspex cages for initiation of rearing colonies and identification of the respective parasitoids. 

### 2.2. Identification of the Parasitoids

#### 2.2.1. Molecular Identification

Ten female parasitoid wasps obtained from the rearing colony were frozen at −20 °C in Eppendorf tubes. To obtain high DNA quality, the heads of the parasitoids were excised before extraction according to the protocol of Livak, [45]. For the polymerase chain reaction (PCR) amplification, a Master Mix solution was prepared by mixing the barcoding primers LCO1490 (5′GGTCAACAAATCATAAAGATATTGG3′), and HCO2198 (5′TAAACTTCAGGGTGACCAAAAAATCA) for arthropods identification [46], DDH2O and hot start. Each sample was prepared with a 48 µL master mix solution in a PCR tube and 2 µL of DNA was added to the solution. The products were then displayed in Proflex 96-well thermal cycler (Applied Biosystems, Waltham, MA, USA) for PCR running. The PCR products were gel extracted, purified, and sent to Macrogen Europe BV (Meibergreef, Amsterdam, The Netherlands) for sequencing. DNA sequences were manually edited in BioEdit version 72.5 [47]. The forward and the reverse sequences were edited pairwise, and the consensus sequences were then generated for each sample. These were then blasted in the National Center for Biotechnology Information (NCBI) and the Barcode of Life Data System (BOLD) databases using nucleotide sequences to determine any similarities with previously described sequences. Nucleic acid sequences from the two samples (herein referred to as first and second samples) were registered in GenBank^®^ as per accession numbers MZ314460, MZ314461, and MZ314460 for the first sample and MZ318061, MZ318062, MZ318063 and MZ318064 for the second sample. Voucher specimens were deposited in the Canadian National Collection of Insects (CNC) at Agriculture and Agri-Food Canada. 

#### 2.2.2. Morphological and Ecological Identification

Morphological identification was performed on specimens that did not provide a high match with any species in NCBI and BOLD. Samples of this specific parasitoid were placed in a refrigerator at −20 ± 2 °C for 5 min to incapacitate them for identification under a reflected stereomicroscope (Leica EZ4D digital stereomicroscope; Leica Microsystems, Heerbrugg, Switzerland). 

### 2.3. Parasitoid Species Effectiveness and Distribution in the Field

The geographical coordinates and altitude were recorded for each farm, where samples were collected, using the Global Positioning System (GPS). The percent parasitism of each parasitoid species on the associated host, *T. absoluta*, was estimated as (the number of parasitoids emerging divided by the total number of *T. absoluta* and the parasitoids from a sample) × 100 [44,48], while the level of *T. absoluta* infestation was recorded as the sum of the number of *T. absoluta* recovered from a sample and the number of recovered parasitoids.

### 2.4. Agroecological Parameters Effect on the Parasitoid Species

In addition to the predators and parasitoids obtained from infested plant material, a survey was conducted to assess the application of pesticides at the tomato farms. Farmers were asked whether they applied pesticides, and if so, the last pesticide application date was recorded. Farms where no pesticide had been applied in the two weeks preceding the survey, were considered as farms with infrequent pesticide application. The tomato production system was also recorded as either an open field or greenhouse/screen house production. Additionally, an average number of *N. tenuis* per kilogram of tomato leaves were estimated and four classes were obtained for this variable 0 = absence of *N. tenuis*, [1,2,3,4,5,6,7,8,9,10,11,12,13,14,15,16,17,18,19,20,21,22,23,24,25,26,27,28,29,30,31,32,33,34,35,36,37,38,39,40,41,42,43,44,45,46,47,48,49,50] = low-level presence of *N. tenuis*, [50,51,52,53,54,55,56,57,58,59,60,61,62,63,64,65,66,67,68,69,70,71,72,73,74,75,76,77,78,79,80,81,82,83,84,85,86,87,88,89,90,91,92,93,94,95,96,97,98,99,100] = medium level presence of *N. tenuis* and [>100] = high presence level of *N. tenuis*. The different factors were then correlated with the parasitism level of each species. 

### 2.5. Prediction of Habitat Suitability

**Occurrence records:** A total of 21 georeferenced points for the first specimen (identified as *B. nigricans* and 23 for the second specimen identified as *S.* sp. nr *japonicus* were obtained from different sources to predict the habitat suitability of both parasitoid species. For *B. nigricans* four points were obtained from Biondi et al. and Gabarra et al. [49,50]; five from the Global Biodiversity Information Facility (GBIF) and 12 from points sampled in the present study. Similarly, for the occurrence records of *S.* sp. nr *japonicus,* four points were obtained from GBIF, eight from the Centre for Agriculture and Bioscience International (CABI) [51], and eight were gathered from published articles [49,50,52,53], and three from the current study.

**Environmental Variables and modeling:** Nineteen bioclimatic variables and elevation data at 2.5-minute spatial resolution were sourced from the WorldClim database [54]. These data points were integrated with occurrence records to predict the habitat suitability of *B. nigricans* and *S*. sp. nr *japonicus* under current climatic conditions. The bioclimatic variables are important in predicting the habitat suitability of the insect species because they reflect diverse characteristics of temperature, precipitation, and seasonality which are factors that affect the distribution and abundance of insects [55]. For future predictions of the distribution of the two parasitoids (i.e., the year 2050), we used simulated bioclimatic variables of representative concentration pathways (RCP 2.6).

### 2.6. Data Analysis

Mega software 10.2.5 [56] was used to generate the phylogenetic tree. The models with the lowest Bayesian Information Criterion (BIC) scores and the lowest Akaike Information Criterion (AIC) value were used to describe the replacement design. All positions with less than 95% site coverage were eliminated. Parameters with fewer than 5% alignment gaps, missing data, and ambiguous bases were allowed at any position (partial deletion option) were used to build the tree. 

*Tuta absoluta* infestation level, as well as parasitoid species abundance, were correlated with the different biotic parameters describing the sites using a general linear model with a *negative binomial distribution*. All analyses were performed in R [57]. 

For parasitoid species ecological niche prediction, a Pearson test applying a threshold correlation coefficient (r) > 0.7 was run in R to check for collinearity among the environmental variables. Only the variables which were not correlated were used in the prediction models using the maximum entropy algorithm (Maxent) developed by Phillips et al. [58], Maxent is one of the most popular modeling tools that has been widely used for predicting the distribution and ecological niche for many insect species [59]. It uses the correlative approach to correlate species occurrence to environmental layers and performs well even with a small number of occurrence records [60]. QGIS [61] was used to show the parasitoid distribution within the study area as well as in their ecological niche model.

## 3. Results

### 3.1. Parasitoid Identification

#### 3.1.1. Molecular Identification

The comparison of the sequence of the first specimen, Accession MZ314460, in NCBI database showed 99.53% genetic similarity (e = 0E00) with the accession MH733585 (Italy) of *Bracon nigricans* Szépligeti (Hymenoptera: Braconidae), 98.28% with MN525195 and 98.12% with MN525197 both unidentified Braconidae from India [53]. Thus, we concluded that the parasitoid species was *B. nigricans* (phylogenetic tree complex is shown in Figure 1).

For the second specimen, the comparison was done both in NCBI and in BOLD. The results from the two databases showed less than 98% similarity; with the highest percentage similarity, 97.66% obtained from sample BIOUG48548-E02 collected in Mpumalanga province, South Africa by Albert Smith. The sequences were also compared with available mitochondrial COI of *Steneomesius* sp. and some closely related sequences in the GenBank. Sequences from specimens of the current study are in the same clade as *Stenomesius Hansson* (Hymenoptera: Eulophidae) from Costa Rica (Figure 2). We concluded that the parasitoid species was in the genera *Stenomesius* but could not identify it to species level using molecular identification. 

#### 3.1.2. Morphological and Ecological Identification

Morphological identification was done for the Eulophidae, *Stenomesius* sp. only, following the description by Reina and La Salle [62]. A comparative analysis of the morphological similarities and differences was done, with the following key features considered: forewing, flagellum, post marginal and stigmal veins, scape, female antennae, mesosoma, setae, and propodeum, as well as the body color. The key features of the *Stenomesius* sp. recovered in the present study include: a scape slightly exceeding the apex of the vertex (Figure 3A), female antennae with a slender scape, and a 4-segmented funicle (Figure 3B). The flagellum has 1–2 anelli (Figure 3B). The forewing has a submarginal vein with more than 4 setae (Figure 3C). The post marginal vein is at least 1.4 times the length of the stigmal vein (Figure 3D). The petiole is not separate (Figure 3E). The propodeum is connected with two submedian carinae in the middle making an H- or X-shaped structure (Figure 3F,G). The complete body is shiny yellowish in color, with a dark spot on the dorsal part of the abdomen and the thorax (Figure 3F,G). These morphological characteristics agree with descriptions for the *Stenomesius* genus.

Three species of the genus *Stenomesius* have been reported in the Afrotropical region, namely *Stenomesius elegantulus* (Risbec) (Hymenoptera: Eulophidae) in Cameroon and Senegal and *S. japonicus* in the Afrotropical, Palaearctic, Indo-Malaya, and Australian regions [63,64] while *S**tenomesius rufescens* (Retzius) (Hymenoptera: Eulophidae) was described for the first time in Africa, in Egypt [64], and this species is supposed to be distributed Nearctic and Palaearctic regions. However, the species has been identified in Kenya [65]. *Stenomesius elegantulus* differs from *S. japonicus* mainly by its relatively shorter scutellum abruptly black compared to yellow axillae [52]. The head of *S. rufescens* is mostly black, the mouth and eye rims are reddish yellow, the two edges of the forewing, the dorsum of the mid-ventrum with the shoulder blades and the shield are ochre yellow, and other parts are black [66]. 

The *Stenomesius* species examined in the current study has a shiny head and body (Figure 3A,F,G). The *S. japonicus* specimens identified by Boucek [52] developed on small size caterpillar hosts, preferentially on herbaceous plants. The parasitoid has been reared on the leaf miner *Stomopteryx nerteria* (Meyrick) (Lepidoptera: Gelechiidae) on groundnut, *Acrocercops* sp. (Lepidoptera: Gracillariidae), and on *Heliothis armigera* (Hübner) (Lepidoptera: Noctuidae). *Stenomesius japonicus* was previously known as a parasitoid of *Liriomyza* spp. (Diptera: Agromyzidae) [67]. A few years after the *T. absoluta* invasion of different parts of the world, the parasitoid *S. japonicus* was recovered from field-collected *T. absoluta* larvae in the northeast of Spain [50] and recently in Syria [68]. Based on all these analyses, we identified this *stenomesius* species as *S.* sp. nr *japonicus*.

### 3.2. Parasitoid Species Effectiveness and Distribution in the Field

The occurrence and diversity of the parasitoids and parasitism rates significantly varied in the 29 sites sampled. *Bracon nigricans* was recorded only in Kirinyaga county (Figure 4B and Table 1) while *S.* sp. nr *japonicus* was recovered in both Nairobi and Kirinyaga counties (Figure 4C and Table 1). However, none of these parasitoid species were recovered from Nakuru County (Figure 4 and Table 1). Between sites, parasitism rates by *S.* sp. nr *japonicus* and *B. nigricans* ranged between, 0 to 17 % and 0 to 21%, respectively (Figure 4B,C). Whereas the infestation level varied from 52 to 1649 (emerged *T. absoluta* and parasitoids) per Kg of infested leaves (Table 1). in the different counties. Additionally, the highest parasitism rate was observed in June and May respectively for *B. nigricans*, and *S*. sp. nr *japonicus* in 2020 (Table 1).

### 3.3. Effect of Agroecological Parameters on Parasitoid Abundance

The low level of *N. tenuis* was significantly positively correlated (*z* value = 3.02, *p* = 0.002) with *B. nigricans* abundance (Table 2), while both insecticide application negatively affected *S.* sp. nr *japonicus* population density (*z* value = −5.56, *p* < 0.001) as well as open-field production (*z* value = −4.27, *p* < 0.001) (Table 2).

### 3.4. Habitat Suitability Prediction

Evaluation of the model showed high accuracy for predicting the habitat suitability of *B. nigricans* with the area under the curve (AUC) = 0.80. Similarly, the model predicted *S.* sp. nr *japonicus* occurrence well with the area under the curve (AUC) = 0.90. Out of the six variables that have been used to predict the habitat suitability of *B. nigricans*: precipitation of driest month (Bio14), precipitation of coldest quarter (Bio19), mean temperature of the driest quarter (Bio9), mean temperature of the wettest quarter (Bio8), and precipitation seasonality (Bio15) (Table 3). Isothermality (Bio3), mean temperature of the driest quarter (Bio9), mean diurnal range (Bio2), precipitation of wettest month (Bio13), precipitation of warmest quarter (Bio18), elevation, mean temperature of the wettest quarter (Bio8), and precipitation of coldest quarter (Bio19) were the bioclimatic variables that contributed to predicting the occurrence of *S.* sp. nr *japonicus* (Table 4). The model prediction showed that most parts of the world are suitable for *B. nigricans* to thrive under both current and future climatic scenarios (Figure 5 and Figure 6). High to very high suitability for *B. nigricans* occurrence is predicted across Africa under current climatic conditions (Figure 5A). In North Africa, the habitat suitability for this species is moderate under future scenarios (Figure 5B). In the current scenario, *S.* sp. nr *japonicus* showed a high to very high suitability to South America, Australia, and some location in southern Asia and southeast Asia (Figure 6A), with the occurrence probability, greatly reduced in the future climatic scenario (Figure 6B). However, the suitability level for this parasitoid is lower in a major part of Africa in both scenarios (Figure 6).

## 4. Discussion

The current study investigated the presence, distribution, and natural parasitism of promising native parasitoids of *T. absoluta,* six years after its invasion into Kenya. This investigation was conducted to determine their potential contribution to biological control of *T. absoluta* as well as for implementation into IPM strategies. We reported two parasitoid species, *B. nigricans,* and *S.* sp. nr *japonicus*. This is not the first report of these species on the African continent [30,69], but it is the first report of the two species in Kenya, although Kinyanjui et al. [65] reported the presence of a *Bracon* sp. and *S. rufescens*.

*Bracon nigricans* is a generalist idiobiont parasitoid of different pests such as *Spodoptera littoralis* Boisduval (Lepidoptera: Noctuidae), *Lobesia botrana* (Denis & Schiffermüller) (Lepidoptera: Tortricidae) [70,71]. On *T. absoluta*, it has been found in various regions such as Western and Southern Europe, Russia, the Middle East, and China [72,73,74,75]. Other *Bracon* species reported on *T. absoluta* are *Bracon lucileae* (Marsh), *B. lulensis,* and *B. tutus* (All) Berta & Colomo [8,72,73]. The maximum *B. nigricans* parasitism rate by *B. nigricans* on *T. absoluta* recorded in this study, was 21%. However, Idriss et al. [30] reported 54% parasitism under laboratory conditions with the highest parasitism obtained on 4th instar larvae compared to only 10% parasitism of 3rd instar larvae. Additionally, a significant pest killing rate, higher than the parasitism rate is observed for this parasitoid species [30,76,77].

*Stenomesius* spp. are generalist ectoparasitoids from several Lepidoptera families, such as Pyralidae, Noctuidae, Gelechiidae, Tortricidae, Lyonetiidae, Glyphipterygidae [78], but also from Diptera pest [79]. *Stenomesius* sp. nr *japonicus* has been associated with *Aproaerema modicella* Deventer (Lepidoptera: Gelechiidae), the groundnut leafminer [53,80] and with *Phyllonorycter* leafminers (Lepidoptera: Gracillariidae) [81] in India and Thailand, and Japan respectively. In Africa, *Stenomesius* species have been identified as parasitoids of *A. modicella* in Uganda and the DRC [82]. On the same host plant, *S. japonicus* attack *Liriomyza* sp. Mik (Diptera: Agromyzidae) [84]. *Stenomesius* sp. near *japonicus* was also recorded from *T. absoluta*-infested tomato plants in Spain and France [51,72,74]. *Stenomesius* sp. has also been reported on *T. absoluta* in Algeria [26], while *S. rufescens* was previously identified in Kenya on the same host [65]. *Stenomesius* sp. nr *japonicus* has been ranked among the three most frequent native parasitoid species recovered from *T. absoluta* in Spain [50]. It is not surprising that we recovered this parasitoid on *T. absoluta*-infested tomato plants in Kenya. 

In the current study, we established up to 17% parasitism rate per site by *S.* sp. nr *japonicus* under field conditions. However, Chailleux et al. [84] found 50% parasitism when offering five hosts to one female of *S.* sp. nr *japonicus*, under laboratory conditions, and 35% field parasitism was reported by Youssef et al. [68]. Additionally, 12 females parasitized around 33% of *T. absoluta* larvae when *S.* sp. nr *japonicus* was exposed alone to *T. absoluta* larvae while only 8% parasitism was recorded when competing with the predator *Macrolophus pygmaeus* Rambur (Hemiptera: Miridae) [85]. A possible explanation for the lower rate of parasitism obtained in this study is that it is from open-field tomatoes where plants were infested with different immature stages, and some 4th instar larvae could have already left the plants to pupate in the soil. Parasitism rates under field conditions will most probably always be lower than laboratory conditions since the pest and parasitoids are not kept under confined conditions. Additionally, the level of infestation, i.e., the host-parasitoid ratio could have affected parasitism of *S*. sp. nr *japonicus*, since, in Table 1, 45% parasitism was observed when the infestation was lowest. 

*Bracon nigricans* is a known larval parasitoid of *T. absoluta* [30,76]. The low-level presence of the generalist, omnivorous mirid *N. tenuis* in the field was positively correlated with *B. nigricans* parasitism while no significant effect on *S.* sp. nr *japonicus* was noticed (Table 1). Similarly, the number of progeny and emerged adults of the endoparasitoid of *T. absoluta*, *Dolichogenidea gelechiidivoris* (Marsh) (Hymenoptera: Braconidae) was not affected by the release of this predator [86]. It is, however, known from several studies that the release of *N. tenuis* in tomato fields, considerably reduces the persistence of *T. absoluta* [87,88], although *N. tenuis* prefers eggs to larvae [89]. However, larval ectoparasitoids go through intraguild predation by mirids in natural habitats [84,90]. Bacci et al. [91] estimated that 57% of *T. absoluta* mortality that occurs under field conditions was during the larval stage, with physiological disorders, parasitism, predation, and entomopathogenic agents as the causes. Third and fourth instar larvae are more susceptible to predatory wasps [91]. In a laboratory study, Chailleux et al. [85] reported that *S*. sp. nr *japonicus* increased from a single couple released in a cage to 60 adults within eight weeks, while its combination with a pair of *M. pygmaeus*, *S.* sp. nr *japonicus* adults decreased by 50 adults for the same duration. 

Application of insecticides in tomato fields was not found to affect the parasitism by *B. nigricans*; however, there was a negative correlation between insecticide application and parasitism by *S.* sp. nr *japonicus* (Table 1). However, Biondi et al. [92] demonstrated that spinosad caused high pupal and adult mortality of 80% and100%, respectively while other bioinsecticides showed no lethal effect on *B. nigricans* but sublethal ones, especially on adult longevity and female fecundity. The impact varies with the active ingredient of the insecticide, combined with biotic conditions [13]. Further, open field tomato farming was found unfavorable to *S.* sp. nr *japonicus*. Similarly, for the congeneric species, *N. nr artynes* recorded more adult emergence (72) in greenhouse production compare to sentinel plants (23 adults) and open field tomatoes (41adults) in Tunisia [27]. Another Eulophid, *Hemiptarsenus varicornis* (Girault) (Hymenoptera: Eulophidae) was promoted as a good candidate for biocontrol of *Liriomyza trifolii* (Burgess) (Diptera: Agromyzidae) due to its ability to effectively perform under greenhouse conditions [93].

The generated Maxent models predicted the presence of *S.* sp. nr *japonicus* and *B. nigricans* in many parts of the world (Figure 5). The widespread distribution of *Stenomesius* sp. as predicted by the Maxent model is validated by the occurrence data of the parasitoid already published [49,50,53]. The model predicted low climatic suitability for *S.* sp. nr *japonicus* in most parts of Africa, in both current and future climatic scenarios. For *B. nigricans*, the Palaearctic region was considered the geographical distribution area [30,50,73]. While from our findings, the Afrotropical, Neotropical, Oriental, and Australasian regions showed better suitability than the Palaearctic region. The previous findings of the habitat suitability of *T. absoluta* in Africa, demonstrated proportional habitat suitability for *B. nigricans* [94,95], a result which juxtaposes very lower habitat suitability observed for *S.* sp. nr *japonicus* in our study. This discrepancy could be explained by the difference in tolerance to environmental factors between *T. absoluta* and the identified parasitoid species. Otherwise, major agricultural regions of Africa are highly suitable for *B. nigricans*. In that regard, *B. nigricans* should be considered for importation into the geographic areas indicated to be suitable for its occurrence. 

## 5. Conclusions

Two native parasitoid species, *B. nigricans* and *S.* sp. nr *japonicus* were reported on *T. absoluta* in Kenya. Differences in parasitism rates occurred and varied between the sites and regions of sample collection, where the highest parasitism by *B. nigricans* and *S.* sp. nr *japonicus* was 21% and 17%, respectively. A low abundance of *N. tenuis* was found to be positively correlated to the occurrence of *B. nigricans*, while pesticide application in either greenhouse or open field tomatoes, did not affect the parasitoid. Open-field cropping, as well as insecticide application, were negatively correlated with parasitism by *S.* sp. nr *japonicus*. The ecological niche prediction for the respective parasitoids indicated a high probability that the potential areas for *B. nigricans* occurrence is in almost all African countries. This result can therefore guide future recovery surveys and the implementation of different biological control strategies against *T. absoluta.* With the estimated performance of these parasitoids under field conditions, where many challenges such as exposure to a wide variety of chemical insecticides and other competitors such as predators are realities, an IPM approach integrating the conservation and/or augmentation of these two parasitoids can be very effective. The combination of the different parasitoid species including the introduced *Dolichogenidea gelechiidivoris* (Marsh) (Hymenoptera: Braconidae) [96,97,98] should also be tested to confirm the theory of diversity of parasitoids for better pest control [99,100,101]. Moreover, the roles of these parasitoids in conservative biological control should be investigated by simulating agro-ecological conditions, and augmentation biological control with a high number of releases should also be tested. 

## Figures and Tables

**Figure 1 insects-13-00496-f001:**
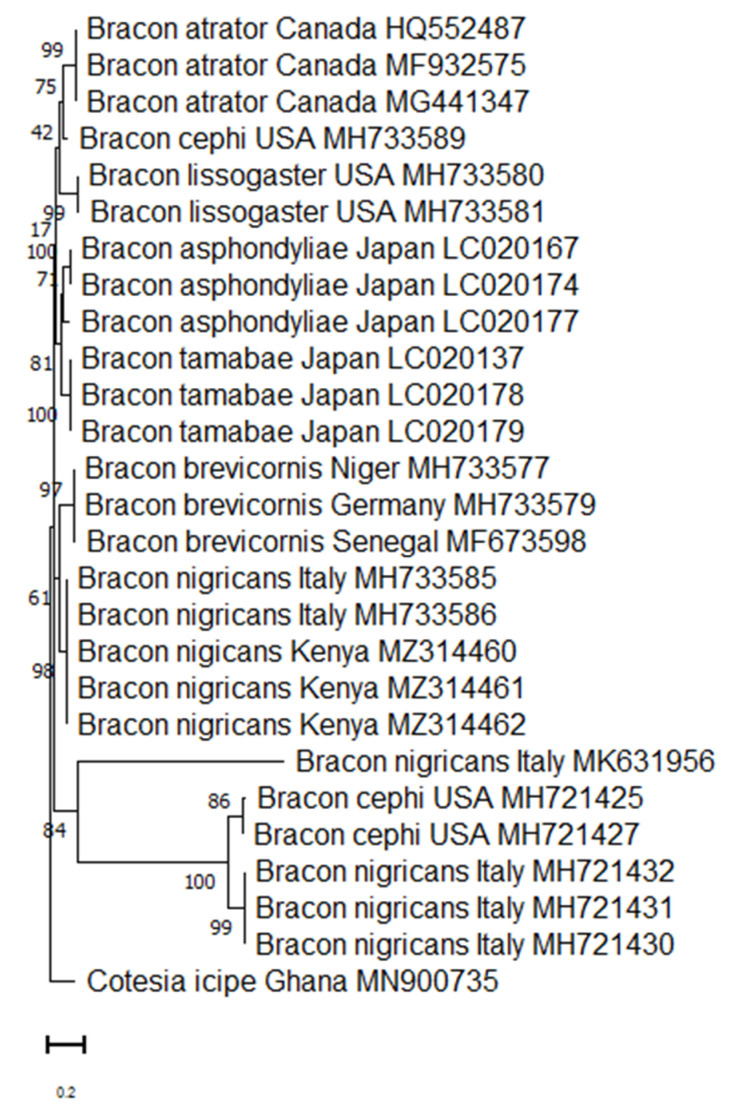
Maximum likelihood phylogenetic tree for the mitochondrial *COI* sequences of *Bracon* species from GenBank together with the specimen *Bracon nigricans* identified in our study.

**Figure 2 insects-13-00496-f002:**
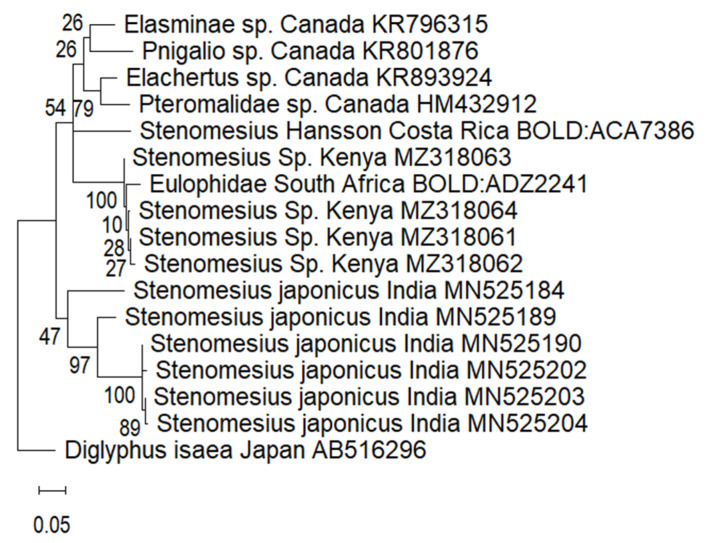
Maximum likelihood phylogenetic tree for the mitochondrial COI sequences of *Stenomesius* species from GenBank and BOLD together with the specimen identified in our study.

**Figure 3 insects-13-00496-f003:**
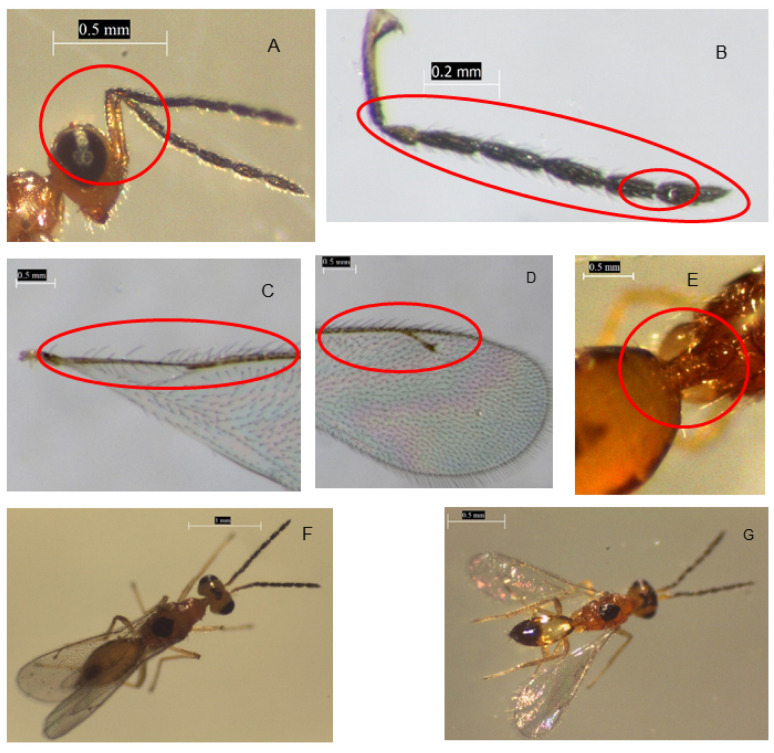
The collected *Stenomesius* sp. near *japonicus* description: (**A**): Scape and vertex, (**B**): Female antenna with scape slender and funicle 4-segmented, (**B**): The flagellum with 1–2 anelli, (**C**): Forewing with submarginal vein with more than 3: setae, (**D**): Postmarginal vein at least 1.5 times the length of the stigmal vein, (**E**): Petiole not separate, (**F**,**G**): Adult with X-shaped structure.

**Figure 4 insects-13-00496-f004:**
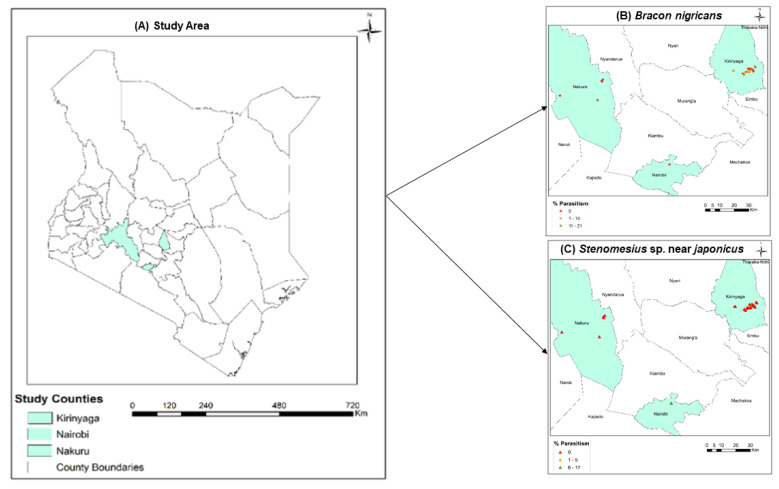
Maps showing (**A**) the study area, (**B**) *Bracon nigricans* parasitism rate, and (**C**) *Stenomesius* sp. nr. *japonicus* parasitism rate, within the sampling sites.

**Figure 5 insects-13-00496-f005:**
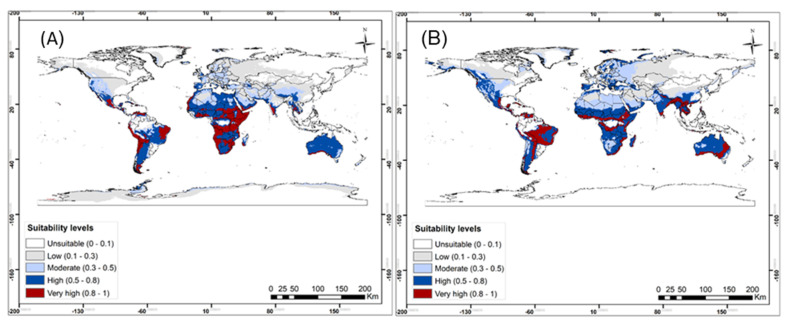
Probable bioclimatic suitability of Bracon nigricans under (**A**) current, and (**B**) representative concentration pathways (RCP2.5) of 2050 climate scenarios.

**Figure 6 insects-13-00496-f006:**
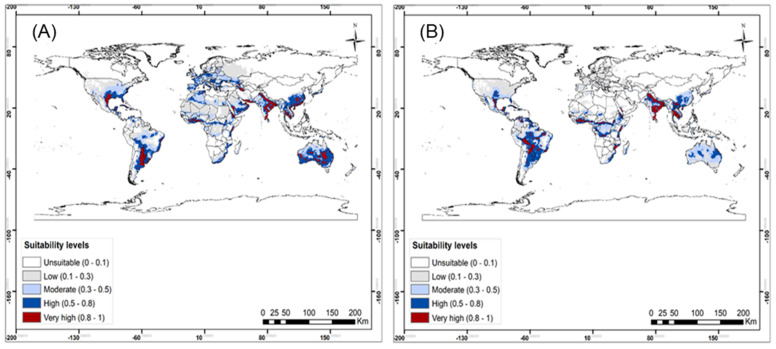
Probable bioclimatic suitability of Stenomesius sp. near japonicus under (**A**) current, and (**B**) representative concentration pathways (RCP2.5) of 2050 climate scenario.

**Table 1 insects-13-00496-t001:** Description of *T. absoluta* emergence per kilogram of infested leaves and the corresponding parasitism rate for each species in the 2020 and 2021 collections.

Years	Months	County	No. of Sites	*T. absoluta* Infestation/kg of Leaves (No.)	*B. nigricans* (%)	*S*. sp. nr *japonicus* (%)
2020	March	Kirinyaga	1	290 ± NA	0	0
	May	Nairobi	1	52 ± NA	0	45.16 ± NA
	June	Kirinyaga	2	230 ± 97	12.02 ± 3.77	0
2021	February	Nairobi	2	267 ± 182	0	25.37 ± 11.96
	March	Nairobi	1	1649 ± NA	0	1.10 ± NA
	March	Nakuru	2	948 ± 542	0	0
	May	Kirinyaga	9	825 ± 146	0.73 ± 0.60	1.21 ± 0.80
	August	Nakuru	2	116 ± 99	0	0
	October	Kirinyaga	12	353 ± 69	5.55 ± 1.86	0.03 ± 0.03
	November	Nakuru	1	241 ± NA	0	0

**Table 2 insects-13-00496-t002:** Statistical estimates for the effect of agroecological parameters on the parasitoid’s abundance.

	Estimate	SE	*z* Value	Pr(>|z|)
*B. nigricans*				
(Intercept)	0.92	0.91	1.02	0.31
Farm with frequent pesticide application	−1.60	0.82	−1.95	0.05
Low level of *N. tenuis*	2.50	0.83	3.02	0.002
Medium level of *N. tenuis*	0.04	1.26	0.03	0.97
High level of *N. tenuis*	1.33	1.23	1.08	0.28
*S.* sp. nr *japonicus*				
(Intercept)	3.06	0.24	12.88	<0.001
Farm with frequent pesticide application	−3.46	0.62	−5.56	<0.001
Open-field production	−3.25	0.76	−4.27	<0.001
Low level of *N. tenuis*	−0.27	1.06	−0.25	0.80
Medium level of *N. tenuis*	0.26	0.39	0.65	0.52
High level of *N. tenuis*	−0.88	0.47	−1.85	0.06

**Table 3 insects-13-00496-t003:** Relative contribution of the various bioclimatic variables for Bracon nigricans ecological niche modeling.

Variables	Percentage Contribution	Permutation Importance
Bio14	40.10	64.00
Bio19	23.26	0
Bio9	14.00	4.10
Bio8	12.70	17.10
Bio15	9.60	14.80
Elevation	0	0

**Table 4 insects-13-00496-t004:** Relative contribution of the various bioclimatic variables for Stenomesius sp. near japonicus ecological niche modeling.

Variables	Percentage Contribution	Permutation Importance
Bio3	57.2	51.9
Bio9	18.0	20.5
Bio2	8.1	4.3
Bio13	6.9	13.0
Bio18	3.9	7.3
Elevation	3.3	0
Bio8	2.4	2.6
Bio19	0.2	0.2
Bio15	0	0

## Data Availability

Data presented in this study are available for public used with the URL: https://dmmg.icipe.org/dataportal/dataset/identification-microhabitat-and-ecological-niche-prediction-of-two-promising-native-parasitoids (15 February 2022).

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
