# Peer review of "Identification, Microhabitat, and Ecological Niche Prediction of Two Promising Native Parasitoids of Tuta absoluta in Kenya"

_insects, 2022, doi:10.3390/insects13060496_

Round 1

Reviewer 1 Report

Dear Autours

Your study reported in this MS aimed to assess the presence, distribution, and efficiency of control of Tuta absoluta in Kenya  by two parasitoid species as well as to determine their suitable habitat for perseverance in biological control programs in  all Africa. This is a very important and interesting question and  it is clear that you have done a lot of field and laboratory work and an interesting mathematical analysis  on the Effect of agro-ecological parameters on parasitoids abundance   but never the less  your manuscript suffers from  some of the major problems that are listed below:

The methods: I do not know if the problem is in the method them self or in the way you have written them, but there is not enough information about the collection of the infested tomato plants who are the basis to all the information you later present. How many replications did you have? how many plants in each farm? , did you return to the same plant again in the next collection or did you sample only once, how often did you do the collection of the leaves, in what dates between March -2020-October 2021.  All this brings to a big lack in the information you claim to present :"Wo present an average of parzitized larvae" but there is to information about the variance (no SE nor SD) and no information about the T. absoluta infestation level each sampling date. When analyzing what effect parasitoids success rate one can not ignore the host's ppulation size.     

Another problem  in the methods lies in the definition of " Farms where no pesticide had been applied in the two weeks preceding survey, were considered as untreated"
But two weeks are not enough time to consider a field as untreated, some pesticides impact stays much longer as you nicely write siting Biondi.

Your main goal was to  assess  the parasitoids efficiency of control of Tuta absoluta but you do nor really present data about the efficacy beside the fact that maximum parasitism rate was 17% and 21% depending on the site, when the range starts at 0% and there is no information about the average and the variance. This does not seem like a very high efficacy rate, and there is no information  about how this rate  affected the T. absoluta  population  along the time. Hence actually the basis of all the nice article you have written is not sound. In the discussion you  mention that Chailleux et al. found 50% parasitism when offering
five host to one female of S. sp. nr japonicus, under laboratory conditions. Additionally, 12 females parasitized around 33% of T. absoluta larvae when S. sp. nr japonicus was exposed alone to T. absoluta larva, but even this is not enough to reduce a pest intrinsic grows rate to go below 1 R<1.

There are also some font size mistakes that should be carefully corrected

Saying all that your MS isnicely written, and has a potential to be an important contribution to future IPM program in Africa, but first the basic faults concerning the materials and methods and the result presented should be corrected.

 I added my comments directly on the MS.

I wish you a very good luck

Author Response

The methods: I do not know if the problem is in the method them self or in the way you have written them, but there is not enough information about the collection of the infested tomato plants who are the basis to all the information you later present. How many replications did you have? how many plants in each farm? , did you return to the same plant again in the next collection or did you sample only once, how often did you do the collection of the leaves, in what dates between March -2020-October 2021.  All this brings to a big lack in the information you claim to present :"Wo present an average of parzitized larvae" but there is to information about the variance (no SE nor SD) and no information about the T. absoluta infestation level each sampling date. When analyzing what effect parasitoids success rate one can not ignore the host's ppulation size.     

Another problem  in the methods lies in the definition of " Farms where no pesticide had been applied in the two weeks preceding survey, were considered as untreated" But two weeks are not enough time to consider a field as untreated, some pesticides impact stays much longer as you nicely write siting Biondi.

Your main goal was to  assess  the parasitoids efficiency of control of Tuta absoluta but you do nor really present data about the efficacy beside the fact that maximum parasitism rate was 17% and 21% depending on the site, when the range starts at 0% and there is no information about the average and the variance. This does not seem like a very high efficacy rate, and there is no information  about how this rate  affected the T. absoluta  population  along the time. Hence actually the basis of all the nice article you have written is not sound. In the discussion you  mention that Chailleux et al. found 50% parasitism when offering five host to one female of S. sp. nr japonicus, under laboratory conditions. Additionally, 12 females parasitized around 33% of T. absoluta larvae when S. sp. nr japonicus was exposed alone to T. absoluta larva, but even this is not enough to reduce a pest intrinsic grows rate to go below 1 R<1.

There are also some font size mistakes that should be carefully corrected

Response :          Thanks for your comments. 29 farms were sampled, several samples were collected ramdomly. The number of samples were between 3 and 12. Only in Nairobi, the same sites were sampled but not during the same production season. More information has been added in the methodology and a new table has been added in the result section (Section 2.1.). Table1 ((Section 3.2.) showing the parasitism depending on the time of sampling in each county.

Point 1: L77: please check font size

Response 1: The font size has been corrected

Point 2: L99-100: There is not enough information about the collection how many replications, if you return to the same plants, dates of collections etc.

Response 2: More details have been added in the methodology (Section 2.1.) and in the result section (Table1) (Section 3.2.).

Point 3: L144-L155: Did you do it only once in each place? this is not enough.

Response 3: No,  several samples were collected randomly in each site. It has been corrected (Section 2.1.).

Point 4: L160-161: Two weeks are not enough time to consider a field as untreated, Some pesticides impact stays much longer

Response 4: It has been corrected and the “untreated” has been replaced by “farms with infrequent pesticide application”

Point 5: L186-189: This should be in the data analysis part

Response 5: It has been moved to the data analysis part

Point 6: L233-234: Font size!

Response 6: it has been corrected

Reviewer 2 Report

In this research article, two efficient, native parasitoids of the invasive tomato pest Tuta absoluta, Stenomesius sp. near japonicus and Bracon nigricans, were identified in Kenya (sub-Saharan Africa) and their abundance, parasitism rates and some of their ecological aspects, such as ecological niche, were studied. Overall, this article, being scientifically sound and well written throughout, is suitable for publication in Insects. I think it can be accepted after making all of the revisions below.

L11-12: you should show here that tomato is the main host for Tuta absoluta (damage is also mainly observed in tomatoes); please consider to rephrase what you said, no need to say "of plants"  and "e.g.", but try to be more direct

L15:  to various insecticide active substances.

L15:   is one of the most

L15:  change "hopeful"  to  "promising"

L16:  we investigated the most efficient indigenous

L17:  ecological nice suitability  ??? niche or nice ??

L20:   in either greenhouse or non-insecticide-treated tomatoes, while

L20:  "non-insecticide treated" tomato fields or what ?? what is the difference between greenhouses and non-insecticide treated tomatoes ??

L21:  in the tomato  ?? which tomato (field or greenhouse ??) ??

L22:  showed that B. nigricans was suitable for

L22:  what did you mean with "larger part of Africa for both current and future scenarios" ?? please be as specific as possible instead of writing such general statements

L25:   delete  "this"

L26:  emerged from T. absoluta were

L34:   the parasitism by S. sp. nr japonicus.

L35:  of parasitoid species

L36:  what did you mean with Africa ? all Africa ?? or which parts (geographical areas) of Africa ?? sub-Saharan Africa ?? North Africa ??  please specify  (similar for L37)

L38-39:  especially B. nigricans could be considered for implementation as a biocontrol agent in Integrated Pest Management programs of T. absoluta in Kenya and other parts of Africa.

L40:  Endogenous ?? did you mean "Indigenous" ?? please correct this

L40-41:   change  "tomato leafminer"  to  "South American tomato pinworm"

L43 (Introduction):  the most up-to-date and relevant review article (Desneux et al. 2022) on the management of Tuta absoluta in all infested areas worldwide is missing in the Introduction. Please add this relevant reference whenever mentioning any corresponding pest management option applied against T. absoluta accross tomato-growing regions around the world.

Citation to add in References:

Desneux, N., Han, P., Mansour, R., Arno, J., Brevault, T., Campos, M.R. et al. Integrated Pest Management of Tuta absoluta: practical implementations across different world regions. J. Pest Sci. 2022, 95,17–39. https:// doi. org/ 10. 1007/ s10340- 021- 01442-8

L46-47:  (Lepidoptera: Gelechiidae)

L49:   to the availability pesticides ??

L77:  (Haliday)

L76-87:   there is a great difference between identifying (a simple discrimination) a parasitoid species parasitizing a pest in a region and performing field releases of the parasitoid in the framework of biological control/IPM programs. In North Africa for example, there have been several efforts of parasitoid releases against T. absoluta, which is missing in this part of your article. Please add here potential info regarding releases of various parasitoid species against T. absoluta in North Africa as reviewed in the relevant up-to-date reference "Mansour and Biondi 2021". The main goal in any IPM strategy is to use these parasitoids in Biological control programs, not to only identify them.

Citation to add:

Mansour, R., Biondi, A. Releasing natural enemies and applying microbial and botanical pesticides for managing Tuta absoluta in the MENA region. Phytoparasitica. 2021, 49, 179–194

L90:  approaches for the management of T. absoluta

L94-95:  to assess the presence and distribution of parasitoid species in open-field and greenhouse tomatoes in Kenya and their effectiveness in controlling T. absoluta as well as to

L107: write species in full with its autorship, order and family for its first mention in the text

L144:  replace "Parasitoid's species efficiency"   with  "Parasitoid species effectiveness"

L150:  change to "N. tenuis"    (species already mentioned in L107)

L152-153:  please indicate according to which scientific reference you are using this formula:  (the number of parasitoids emerging divided by the total number of T. absoluta and the parasitoids from a sample) x100   ???

L156:  on the parasitoid species

L171: add the corresponding names of authors before "[38,39]"

From L195 to L204:  the words should not be italicized; only headings (e.g.  "2.6. Data analysis") should be italicized

L206:  3.1. Parasitoid identification

L210:  (Zappala, 2018,NCBI) ??

L233:  write the name of the author before "[50]"

L246:  Three species of the genus Stenomesius

L258:  by Boucek [41] developed on

L266:  We identified this stenomesius species most probably as S. sp. nr japonicus.

L274:  3.2. Parasitoid species effictiveness and

Figure 4:  this figure should be of a better quality for clarity

L304:   on parasitoid abundance

L305:  positively correlated

L309:  move this table to the next page

L331-332:  the word size for the caption is not adequate, please correct it

L337-338:  check and correct the word size for the figure caption

L341-342:   check and correct the word size for the figure caption

L434:  did not have the parasitoid ??? please correct this

L439:  different biological control strategies against T. absoluta

L445-446:   somthing is wrong in this sentence, please rephrase "Moreover, the roles of these parasitoids in conservative biological control, 445 agro-ecological conditions should be simulated, and"

Author Response

Comments and Suggestions for Authors In this research article, two efficient, native parasitoids of the invasive tomato pest Tuta absoluta, Stenomesius sp. near japonicus and Bracon nigricans, were identified in Kenya (subSaharan Africa) and their abundance, parasitism rates and some of their ecological aspects, such as ecological niche, were studied. Overall, this article, being scientifically sound and well written throughout, is suitable for publication in Insects. I think it can be accepted after making all of the revisions below.

Point 1: L11-12: you should show here that tomato is the main host for Tuta absoluta (damage is also mainly observed in tomatoes); please consider to rephrase what you said, no need to say "of plants" and "e.g.", but try to be more direct

Response 1: The sentense has been reviewed and becomes “Since the arrival of Tuta absoluta, a multivoltine insect species whose larvae develop in leaves, fruits, flowers, buds, and stems of in the word, tomato producers are facing one of its biggest production challenge.”

Point 2: L15: to various insecticide active substances.

Response 2: The word “substances” has been added

Point 3: L15: is one of the most

Response 3: The word “the” has been deleted

Point 4: L15: change "hopeful" to "promising"

Response 4: "hopeful" has been changed to "promising"

Point 5: L16: we investigated the most efficient indigenous

Response 5: The word “on” has been deleted

Point 6: L17: ecological nice suitability ??? niche or nice ??

Response 6: Sorry it has been corrected

Point 7: L20: in either greenhouse or non-insecticide-treated tomatoes, while

Response 7:  Both parameters (production in a greenhouse condition and insecticide application) independantly affected the parasitoid abundance. It has been corrected, it should be “in greenhouse tomatoes and non-insecticide-treated tomatoes”

Point 8: L20: "non-insecticide treated" tomato fields or what ?? what is the difference between greenhouses and non-insecticide treated tomatoes ??

Response 8: Yes greenhouse can be with or without insecticide.

Point 9: L21: in the tomato ?? which tomato (field or greenhouse ??) ??

Response 9: The word “field” has been added since B. nigricans is only recorded in open field

Point 10: L22: showed that B. nigricans was suitable for

Response 10: The words “that” and “was” have been added

Point 11: L22: what did you mean with "larger part of Africa for both current and future scenarios" ??

please be as specific as possible instead of writing such general statements

Response 11: It has been corrected, the sentence becomes “The ecological niche of these two species showed that B. nigricans was suitable for sub-Saharan Africa, a big part of South America and Australia a larger part of Africa for both current and future scenarios.”

Point12: L25: delete "this"

Response 12:  The word “this” has been deleted

Point 13: L26: emerged from T. absoluta were

Response 13: It has been corrected

Point 14: L34: the parasitism by S. sp. nr japonicus.

Response 14: The word “of” has been replaced by “by”

Point 15: L35: of parasitoid species

Response 15: It has been corrected

Point 16: L36: what did you mean with Africa ? all Africa ?? or which parts (geographical areas) of Africa ?? sub-Saharan Africa ?? North Africa ?? please specify (similar for L37)

Response 16: It has been corrected and more precisions have been added

Point 17: L38-39: especially B. nigricans could be considered for implementation as a biocontrol agent in Integrated Pest Management programs of T. absoluta in Kenya and other parts of Africa.

Response 17: Thanks, the sentence has been reviewed as suggested

Point 18: L40: Endogenous ?? did you mean "Indigenous" ?? please correct this

Response 18: Kindly, it should be “indigenous” it has been corrected

Point 19: L40-41: change "tomato leafminer" to "South American tomato pinworm"

Response 19: "tomato leafminer" has been chenged to "South American tomato pinworm"

Point 20: L43 (Introduction): the most up-to-date and relevant review article (Desneux et al. 2022) on the management of Tuta absoluta in all infested areas worldwide is missing in the Introduction. Please add this relevant reference whenever mentioning any corresponding pest management option applied against T. absoluta accross tomato-growing regions around the world. Citation to add in References: Desneux, N., Han, P., Mansour, R., Arno, J., Brevault, T., Campos, M.R. et al. Integrated Pest Management of Tuta absoluta: practical implementations across different world regions. J. Pest Sci. 2022, 95,17–39. https:// doi. org/ 10. 1007/ s10340- 021- 01442-8

Response 20: Thanks, the specific reference has been added ([14]) in “Biological control is recommended for the control of T. absoluta albeit integrated with other control methods [7,14,15]”

Point 21: L46-47: (Lepidoptera: Gelechiidae)

Response 21: It has been corrected

Point 22: L49: to the availability pesticides ??

Response 22: It has been deleted

Point 23: L77: (Haliday)

Response 23: It has been corrected

Point 24: L76-87: there is a great difference between identifying (a simple discrimination) a parasitoid species parasitizing a pest in a region and performing field releases of the parasitoid in the framework of biological control/IPM programs. In North Africa for example, there have been several efforts of parasitoid releases against T. absoluta, which is missing in this part of your article. Please add here potential info regarding releases of various parasitoid species against T. absoluta in North Africa as reviewed in the relevant up-to-date reference "Mansour and Biondi 2021". The main goal in any IPM strategy is to use these parasitoids in Biological control programs, not to only identify them. Citation to add: Mansour, R., Biondi, A. Releasing natural enemies and applying microbial and botanical pesticides for managing Tuta absoluta in the MENA region. Phytoparasitica. 2021, 49, 179–194

Response 24: A sentence has been added and references have been added.

Point 25: L90: approaches for the management of T. absoluta

Response 25: The word “in” has been replace by “for”

Point 26: L94-95: to assess the presence and distribution of parasitoid species in open-field and greenhouse tomatoes in Kenya and their effectiveness in controlling T. absoluta as well as to

Response 26: The sentence has been corrected as suggested

Point 27: L107: write species in full with its autorship, order and family for its first mention in the text

Response 27:  It has been corrected

Point 28: L144: replace "Parasitoid's species efficiency" with "Parasitoid species effectiveness"

Response 28: "Parasitoid's species efficiency" has been replaced by "Parasitoid species effectiveness"

Point 29: L150: change to "N. tenuis" (species already mentioned in L107)

Response 29: It has been corrected

Point 30: L152-153: please indicate according to which scientific reference you are using this formula: (the number of parasitoids emerging divided by the total number of T. absoluta and the parasitoids from a sample) x100 ???

Response 30: Two references have been added

Point 31: L156: on the parasitoid species

Response 31: It has been corrected

Point 32: L171: add the corresponding names of authors before "[46,47]"

Response 32: Authors (Biondi et al. and Gabarra et al.) are now added

Point 33: From L195 to L204: the words should not be italicized; only headings (e.g. "2.6. Data analysis") should be italicized

Response 33: It has been corrected

Point 34: L206: 3.1. Parasitoid identification

Response 34: It has been corrected

Point 35: L210: (Zappala, 2018,NCBI) ??

Response 35: In fact, the sequence was submitted in 2015 by Lucia Zappala. It has been corrected with (NCBI, https://www.ncbi.nlm.nih.gov/nuccore/MH733585)

Point 36: L233: write the name of the author before "[50]"

Response 36: Authors names (Reina and La Salle) has been added

Point 37: L246: Three species of the genus Stenomesius

Response 37: It has been corrected

Point 38: L258: by Boucek [41] developed on

Response 38: It has been corrected

Point 39: L266: We identified this stenomesius species most probably as S. sp. nr japonicus.

Response 39: It has been corrected as suggested

Point 40: L274: 3.2. Parasitoid species effictiveness and

Response 40: It has been corrected

Point 41: Figure 4: this figure should be of a better quality for clarity

Response 41: The clarity of the figure has been improved

Point 42: L304: on parasitoid abundance

Response 42: It has been corrected

Point 43: L305: positively correlated

Response 43: It has been corrected

Point 44: L309: move this table to the next page

Response 44: It has been corrected

Point 45:: L331-332: the word size for the caption is not adequate, please correct it

Response 45: It has been corrected

Point 46: L337-338: check and correct the word size for the figure caption

Response 46: It has been corrected

Point 47:: L341-342: check and correct the word size for the figure caption

Response 47: It has been corrected

Point 48: L434: did not have the parasitoid ??? please correct this

Response 48: It has been corrected, it should be “did not affect the parasitoid”

Point 49: L439: different biological control strategies against T. absoluta

Response 49: It has been corrected, the word “management” has been replaced by “control”

Point 50: L445-446: somthing is wrong in this sentence, please rephrase "Moreover, the roles of these parasitoids in conservative biological control, 445 agro-ecological conditions should be simulated, and"

Response 50: Sorry, the sentence has been reviewed and becomes “Moreover, the roles of these parasitoids in conservative biological control should be investigated by simulating agro-ecological conditions, and augmentation biological control with high number releases should also be tested.”

Reviewer 3 Report

Comments are in Peer review 17932264v1. pdf

Author Response

Point 1: I think it is an interesting study, but the explanations especially those of Materials and Methods and Results should be reviewed. It is not easy to understand them, sometimes it's not easy to follow ideas. Although there is supplementary material some abbreviations may be unknown, for example RCP 26, AUC, Bio…. I think they should be included in the text to make easy to read the paper.

Response 1: The full names of the abbreviation have been added

Point 2: The names of Figures and Tables need to be improved. In Figure 3 you must cite that they belong to Stenomesius sp. It is just an example; you should review all of them.

Response 2: The names of figures and tables have been reviewed

Point 3: In Molecular Identification, thermal profiles must quote the temperature and number of cycles well. You say “ (1) initial denaturation at 95 °C (15 to 40 min); (2) denaturation at 95 °C (repeated 15-40 times)”. It is 15, 30 or 40 minutes? It is needed to know the exact number.

Response 3: Since this is a standard procedure, the sentence has been removed

Point 4:  Only the first time a species is cited should be written order and family to which it belongs, for example Nesidiocoris tenuis Reuter (Hemiptera: Miridae), then no longer. In the example cited, it is repeated on lines 28 and 150. Other species when they are first cited no order or family is cited.

Response 4: It has been corrected

Point 5: I also think it is necessary to quote the authors, for example the in the L 232 you say: "following the description by [50]". You should write Reina et al (50). Please, review all the text.

Response 5: The author's name have been added

Reviewer 4 Report

The  authors investigated "identification, microhabitat and ecological niche prediction of two promosing native parasitoids of Tuta absoluta in Kenya". They identified two native parasitoids of Tuta absoluta using DNA Barcoding technique and further conducted a habitat suitability prediction for the identified parasiotiods. The results seems to be interested to some extent. However, these two parasitoids had been reported in many other African countries, even though they were firstly reported in Kenya, so the novelty of this research was very low. According to these, I think this work is very difficult to be accepted by Insects and suggest to submit to a local journal, such as African Entomology.

Author Response

Response to Reviewer 4 Comments

Point: The  authors investigated "identification, microhabitat and ecological niche prediction of two promosing native parasitoids of Tuta absoluta in Kenya". They identified two native parasitoids of Tuta absoluta using DNA Barcoding technique and further conducted a habitat suitability prediction for the identified parasiotiods. The results seems to be interested to some extent. However, these two parasitoids had been reported in many other African countries, even though they were firstly reported in Kenya, so the novelty of this research was very low. According to these, I think this work is very difficult to be accepted by Insects and suggest to submit to a local journal, such as African Entomology.

Response : Thank you for your comments, however we feel that our work is suitable for a wider readership beyond the local limits as you have suggested.

Round 2

Reviewer 1 Report

I went over the revised manuscript and I was glad to find the authors corrected the issues that needed correction and therefore I think the MS can be published without any further delay.

Author Response

Thank you for your kind response

Reviewer 4 Report

These two parasitoids had been reported in many other African countries, even though they were firstly reported in Kenya, so the novelty of this research was very low. According to these, I think this work is very difficult to be accepted by Insects and suggest to submit to a local journal, such as African Entomology.

Author Response

Point : These two parasitoids had been reported in many other African countries, even though they were firstly reported in Kenya, so the novelty of this research was very low. According to these, I think this work is very difficult to be accepted by Insects and suggest to submit to a local journal, such as African Entomology.

Response : Dear reviewer, it is true that our study is not the first report of Bracon nigricans (Szépligeti) (Hymenoptera: Braconidae) and Stenomesius sp. near japonicus (Ashmead) (Hymenoptera: Eulophidae) as parasitoids of Tuta absoluta (Meyrick) (Lepidoptera: Gelechidae) in Kenya. However, our study is the first attempt to describe these species' preferential microhabitats in different cropping systems. We also predicted their ecological niche around the world. We believe that this result can additionally be useful in developing different approaches of biological control.
